# HARNESSING SPECTRAL REPRESENTATIONS FOR SUBGRAPH ALIGNMENT

## ABSTRACT

With the rise and advent of graph learning techniques, graph data has become ubiquitous. However, while several efforts are being devoted to the design of new convolutional architectures, pooling or positional encoding schemes, less effort is being spent on problems involving maps between (possibly very large) graphs, such as signal transfer, graph isomorphism and subgraph correspondence. With this paper, we anticipate the need for a convenient framework to deal with such problems, and focus in particular on the challenging subgraph alignment scenario. We claim that, first and foremost, the *representation* of a map plays a central role on how these problems should be modeled. Taking the hint from recent work in geometry processing, we propose the adoption of a spectral representation for maps that is compact, easy to compute, robust to topological changes, easy to plug into existing pipelines, and is especially effective for subgraph alignment problems. We report for the first time a surprising phenomenon where the partiality associated to the subgraph is manifested as a special structure of the map coefficients, even in the absence of exact subgraph isomorphism, and which is consistently observed over different families of graphs up to several thousand nodes.

## 1 INTRODUCTION

The ability to align data is at the heart of many successful techniques in machine learning and related areas. In its most abstract form, the problem has a straightforward formulation: Given two generic domains $D_1$ and $D_2$, find a transformation $T$ such that $TD_1 \approx D_2$ according to some approximation metric that depends on the task. Examples of such problems are found in numerous applications, including molecular docking (Gainza et al., 2020), image-based rendering (Fachada et al., 2021) , 3D reconstruction (Zhao et al., 2022), generative models (Dai & Hang, 2021) and style transfer (Zhang et al., 2022), in addition to countless others. Recent remarkable examples include CLIP Meila & Zhang (2021), where images are associated to corresponding captions by aligning their learned embeddings, or MaSIF (Gainza et al., 2020), where the interaction site between protein structures (i.e., the surface patches where the proteins geometrically align) is predicted by a geometric deep learning pipeline.

Perhaps the most challenging setting for alignment problems arises whenever the two domains only correspond *partially*, for example due to the lack of observations or noise in the data. In this case, one is not only interested in aligning the two domains, but also in discovering which portions of the domains actually align. The problem is particularly hard if an exact alignment does not even exist, requiring additional robustness to local perturbations in the data.

In this paper, we focus on the general problem of subgraph alignment, as it is representative of a broad spectrum of applications including those mentioned above. We assume to be given two graphs $G_1$ and $G_2$, where $G_2$ appears within $G_1$, possibly up to topological changes. A special case appears when $G_2$ is isomorphic to a subgraph of $G_1$, which is referred to as subgraph isomorphism (see (ii) in Figure 1). This case is included in our treatment, but we also consider noisier settings where a subgraph isomorphism does not exist (see (iii) in Figure 1), yet a semantic correspondence can still be defined.

**Contribution.** In this paper, we focus in particular on the choice of a *representation* for the correspondence. That is, instead of introducing a new matching pipeline to solve subgraph alignment, we

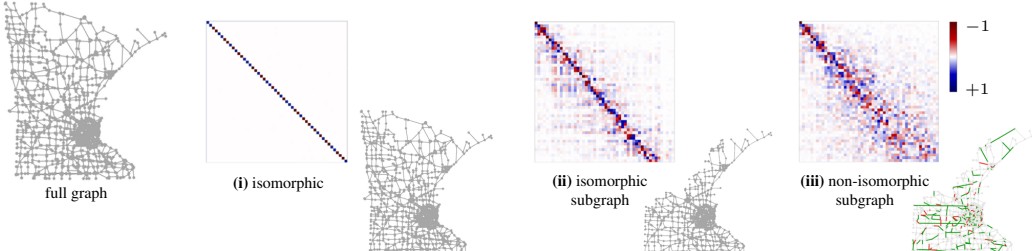

Figure 1: Functional maps of size $50 \times 50$ between a full graph (depicted on the left) and three different graphs, respectively: an isomorphic graph (**i**), an isomorphic subgraph containing $80\%$ of the original nodes (**ii**), and a *non*-isomorphic subgraph obtained by randomly rewiring the former (**iii**). The green edges are novel and randomly added ($10\%$ of the total), the red edges are randomly removed ($10\%$ of the total). The map representation still maintains a visible structure despite the significant changes of connectivity that span several hops.

show an alternative way of representing maps between a graph and its subgraphs. In cases where the map is unknown and must be sought for, the new representation makes the inference problem easier to solve; while if the map is given, the new representation is more compact, has a regularizing effect, and bears a natural structure that is missing from classical representations such as node-to-node binary correspondence matrices.

From a technical perspective, the map representation is defined with respect to a spectral basis; namely, the eigenvectors of the graph Laplacian. This idea, introduced a decade ago in the geometry processing area (Ovsjanikov et al., 2012), brought significant progress to several tasks in graphics and vision – yet, its application to graphs has been largely overlooked.

We claim that part of the reason is a common misconception. The lack of a smooth metric (i.e., a smooth manifold underlying the graph) leads to the assumption that key properties of the spectral representation of the maps, such as those observed in Ovsjanikov et al. (2012); Rodolà et al. (2017), only exist for surface domains. With this work, we challenge this view by showing extensive empirical evidence that not only these spectral maps are applicable to general subgraphs, but also that they exhibit robustness properties that go beyond what was shown on surfaces. Therefore, we propose to embrace the spectral representation of maps as compact, efficient, interpretable, robust, and easy to manipulate objects that can be naturally integrated into several pipelines, including but not limited to graph learning models.

We summarize our main contributions as follows:

- We propose the adoption of spectral representations for maps between graphs and *subgraphs*. For the first time, we show that such maps exhibit a special structure in their coefficients, capturing the similarity between the Laplacian eigenspaces of the two graphs.
- We further show robustness of the representation to topological modifications of the subgraph, due for example to graph rewiring. This leads to well-defined maps even in the absence of exact isomorphism.
- We include extensive experiments showing practical applications, such as signal transfer and subgraph matching, on graphs spanning a few dozen to tens of thousands of nodes, and demonstrate key benefits in terms of robustness to noise, interpretability, and computational complexity.

All the code and generated data will be publicly released upon acceptance.

## 2 RELATED WORK

Graph alignment problems are ubiquitous in applications from social network analysis (Liu et al., 2016) to bioinformatics (Singh et al., 2008). Given its relevance, a rich body of literature is devoted to this problem. A comparative study on several network alignment techniques can be found in Trung et al. (2020). Though not exhaustive, we discuss the most relevant works in the following.

**Node-to-node correspondence.** IsoRank (Singh et al., 2008) is widely considered the baseline for protein-to-protein interaction (PPI) alignment. It constructs an eigenvalue problem for every pair of input networks and extracts a global alignment across a set of networks by a $k$-partite matching. Subsequently, it inspired other works, such as Liao et al. (2009); Nassar et al. (2018); Feizi et al. (2019), that still use a spectral approach. Other works leveraged an optimization process based on attribute and topological information (Zhang & Tong, 2016) or propagated the ground-truth matches over the input graphs (Yartseva & Grossglauser, 2013; Kazemi et al., 2015). Another thriving line of works first computes node embeddings and then aligns the nodes using the similarity between these features. The embeddings can be computed from the Laplacian eigenvectors (Knossow et al., 2009), the connectivity structure and node attributes (Heimann et al., 2018) or directly use representation learning methods (Liu et al., 2016; Man et al., 2016; Zhou et al., 2018), thus requiring manually corresponding matches during the training phase.

All these methods directly look for a node-wise correspondence between input graphs, soon becoming infeasible when the size of the graphs reaches thousands of nodes. We investigate a different approach, where we adopt a *functional* (as opposed to node-wise) representation to define a correspondence between graphs. Furthermore, previous works mostly focus on a single application, either global PPI network alignment (Singh et al., 2008; Liao et al., 2009), synthetic datasets with structural noise (where a small portion of edges are randomly removed) (Hermanns et al., 2021; Trung et al., 2020), or social networks (Liu et al., 2016; Man et al., 2016; Zhou et al., 2018). In our analysis, we address the general task of *subgraph alignment*, where a large portion of the graph is missing, without focusing on any specific domain. This is also different from the problem of subgraph isomorphism, which concerns the decision problem as to whether small query graphs exist within larger graphs (Duong et al., 2021).

**Functional correspondence.** The functional maps framework (Ovsjanikov et al., 2012) was first introduced in the shape analysis field to find correspondences between deformable 3D shapes. Thanks to its flexibility, many extensions of this framework were later proposed, improving the correspondence accuracy by means of dedicated regularizers (Ovsjanikov et al., 2017; Nogneng & Ovsjanikov, 2017; Ezuz & Ben-Chen, 2017; Melzi et al., 2019; Ren et al., 2020a). To allow matching non-isometric shapes, Kovnatsky et al. (2013) apply the framework to approximate eigenbases obtained with a joint diagonalization algorithm. The same approach was later adapted to *partial* shape matching in Rodolà et al. (2017); Litany et al. (2017); Cosmo et al. (2016c), where the goal is to map a part of a deformed 3D shape to a possibly incomplete model.

In the context of graphs, a first attempt to represent similarity between graphs in a functional space was proposed in Wang et al. (2019). However, there the functional representation is on the edge domain, and its application is limited to the case of Euclidean graphs. More related to our work is the recent GRASP (Hermanns et al., 2021), which detects an alignment among graphs by employing a functional correspondence among pre-aligned Laplacian eigenvectors. Different to our work, GRASP considers only the setting of noisy complete graphs, i.e., a full network is perturbed by randomly deleting edges with a probability $p$ up to 0.25. Instead, we study the correspondence problem on a much broader class of graphs, undergoing strong partiality transformations in addition to strong perturbation of the connectivity, and consider larger scales reaching up to several thousand nodes.

## 3 PRELIMINARIES

**Graphs and Laplacian eigenvectors.** We consider undirected, unweighted graphs $G = (V, E)$ with nodes $V$ and edges $E \subseteq V \times V$. We denote as $A \in \{0, 1\}^{|V| \times |V|}$ the adjacency matrix of $G$, which is a binary matrix where $A(i, j) = 1$ if an edge connects node $i$ to node $j$, and $A(i, j) = 0$ otherwise.

The symmetric normalized Laplacian for $G$ is defined as the square matrix $\mathcal{L} = I - D^{-\frac{1}{2}} A D^{-\frac{1}{2}}$, where $D$ is a diagonal matrix of node degrees, with entries $D(i, i) = \sum_{j=1}^{|V|} A(i, j)$. This linear operator is symmetric and positive semi-definite; it admits an eigendecomposition $\mathcal{L} = \Phi \Lambda \Phi^\top$, where $\Lambda$ is a diagonal matrix that contains the eigenvalues, and $\Phi$ is a matrix having as columns the corresponding eigenvectors. Throughout this paper, we assume the eigenvalues (and corresponding

eigenvectors) to be sorted in non-descending order $0 = \lambda_1 \leq \lambda_2 \leq \ldots \leq 2$; this assumption is important for interpreting the functional maps that we define in the sequel.

Each eigenvector $\phi_l$ for $l = 1, \ldots, |V|$ has length $|V|$, and can be interpreted as a scalar function defined on the nodes of the graph; for this reason, we will occasionally refer to them as eigen*functions* for added clarity. The eigenfunctions form an orthonormal basis for the space of functions defined on the graph nodes. Usually, one may consider a subset of eigenfunctions, namely those associated with the $k$ smallest eigenvalues, to approximate the graph signals in a compact way.

**Functional maps.** Ours is a direct adaptation of the functional map representation introduced in Ovsjanikov et al. (2012) for pairs of quasi-isometric surfaces. We present the original framework for surfaces here, and discuss its application to graphs and subgraphs in Sections 4 and Appendix A.

Consider two smooth manifolds $\mathcal{M}$ and $\mathcal{N}$, and let $T : \mathcal{N} \rightarrow \mathcal{M}$ be a point-to-point map between them. Given a scalar function $f : \mathcal{M} \rightarrow \mathbb{R}$, the map $T$ induces a functional mapping via the composition $g = f \circ T$, which can be seen as a linear map $T_F : f \mapsto g$ from the space of functions on $\mathcal{M}$ to the space of functions on $\mathcal{N}$. As a linear map, the functional $T_F$ admits a matrix representation after choosing a basis for the two function spaces.

To this end, consider a discretization of $\mathcal{M}$ and $\mathcal{N}$, with vetices $V_1$ and $V_2$ respectively, and the corresponding disretized version of their Laplace-Beltrami operators (LBOs) (the counterpart of the graph Laplacian on smooth manifolds). The first $k$ eigenfunctions of the two LBOs can be concatenated side by side as columns to form the matrices $\Phi \in \mathbb{R}^{|V_1| \times k}$ and $\Psi \in \mathbb{R}^{|V_2| \times k}$. Further, assume the pointwise map $T$ is available and encoded in a binary matrix $S$, such that $S(y, x) = 1$ if $y \in V_2$ corresponds to $x \in V_1$, and 0 otherwise. By choosing $\Phi$ and $\Psi$ as bases, the functional map $T_F$ can be encoded in a small $k \times k$ matrix $C$ via the change of basis formula:

$$C = \Psi^\dagger S \Phi \,, \tag{1}$$

where $\dagger$ is the Moore-Penrose pseudoinverse. The size of $C$ does *not* depend on the number of points in $\mathcal{M}$ and $\mathcal{N}$, but only on the number $k$ of basis functions. In other words, $C$ represents the linear transformation that maps the coefficients of any given function $f : \mathcal{M} \rightarrow \mathbb{R}$ expressed in the eigenbasis $\Phi$, to coefficients of a corresponding function $g : \mathcal{N} \rightarrow \mathbb{R}$ expressed in the eigenbasis $\Psi$.

When the pointwise similarity $S$ is unknown, one can directly compute the matrix $C$ as the solution of a regularized least-squares problem with $k^2$ unknowns, given some input features on the two surfaces (e.g., landmark matches or local descriptors). For further details we refer to Ovsjanikov et al. (2012; 2017).

## 4    FUNCTIONAL MAPS FOR SUBGRAPHS

In this work, we consider the setting where we are given a graph $G_1$ and a possibly noisy subgraph $G_2 = (V_2, E_2)$ of $G_1$, such that $V_2 \subseteq V_1$ and $E_2 \subseteq E_1$. When moving from surfaces to graphs, Equation 1 takes a simpler expression as we show in Appendix A.1.

### 4.1    MOTIVATION

Classically, maps are represented as binary matrices $S$ whose dimensions scale quadratically with the number of nodes in the graphs. One can directly adopt Equation (4) to shift to a spectral representation $C$ of the map; however, recall that we consider the setting where $G_2$ is a *subgraph* of $G_1$. This simple fact leads to the following important observation, that is central to our contribution:

> In many practical cases, the eigenspaces of the normalized graph Laplacian are well preserved under *non-isomorphic* transformations of the graph, including strong partiality, topological perturbations, and edge rewiring.

Put simply, the values of the Laplacian eigenfunctions stay approximately the same (up to sign, in case of simple spectrum) at the nodes that are not directly involved in the perturbation – which is to say that the eigenvectors of the partial graph $G_2$, encoded in $\Phi_2$, correlate strongly with the those of $G_1$, encoded in $\Phi_1$. This observation is not trivial and has not been reported before, to the best of our knowledge.

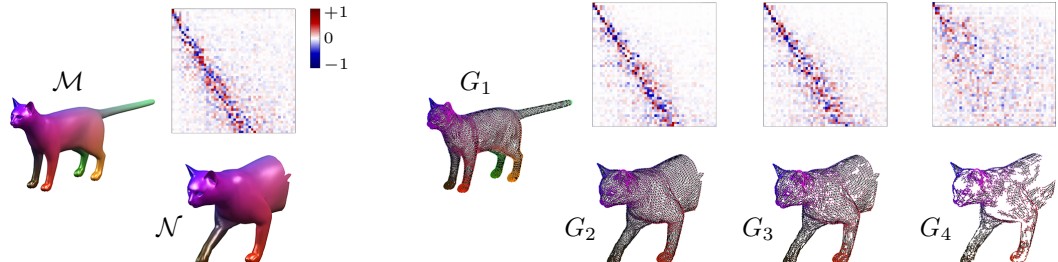

Figure 3: *Left*: Functional map matrix between a smooth surface $\mathcal{M}$ and a deformed part $\mathcal{N}$; the slanted-diagonal structure suggests that the eigenspaces of $\mathcal{M}$ are mostly preserved in $\mathcal{N}$. *Right*: Functional map matrices between a graph $G_1$ and different subgraphs; $G_2$ is obtained by removing 40% of the nodes of $G_1$, while $G_3, G_4$ are obtained by removing 55% and 80% of the edges from $G_2$ respectively. The slanted-diagonal structure can still be observed, and gets dispersed only at very high partiality. In the graphs above, corresponding nodes have the same color.

In the sequel we show extensive empirical evidence of this surprising behavior, and describe its practical consequences.

## 4.2 EIGENVECTORS CORRELATION

To get a better understanding of this phenomenon, in the inset we show an example where the Laplacian eigenfunctions of a Minnesota subgraph strongly correlate with the eigenfunctions of the complete graph, i.e., the Laplacian eigenfunctions have similar values at corresponding nodes, up to sign. Above each image, we also report the index of the plotted eigenfunction, leading to the following remark:

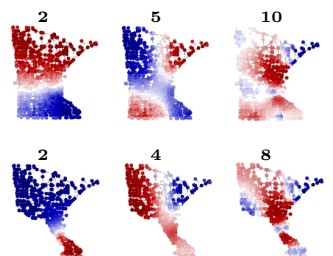

Figure 2: Corresponding eigenfunctions of Minnesota (top row) and its subgraph (bottom row).

***Remark (eigenvector indexing):*** The eigenfunctions of the complete graph and those of the subgraph do not necessarily correlate at the same index (see pair 5-4) and the correlation may not be exact (see pair 10-8); the extent to which the eigenfunctions correlate is captured precisely by the structure of $C$.

In 3D geometry processing, a similar behavior was observed for the discrete Laplace-Beltrami operator under partiality transformations (Rodolà et al., 2017; Postolache et al., 2020); however, their theoretical analysis assumes the data to be Riemannian surfaces with a smooth metric – an assumption that does *not* hold in the case of general graphs. We refer to Appendix A.2 for further details.

## 4.3 MAP STRUCTURE

The most direct consequence of this preservation of eigenspaces is reflected in the structure of the functional map $C$. According to Equation (4), each coefficient $c_{ij}$ of $C$ corresponds to a dot product between $\phi_i^2$ and $S\phi_j^1$; this measures the correlation *at corresponding nodes* between a Laplacian eigenvector $\phi_i^2$ of $G_2$, and a Laplacian eigenvector $\phi_j^1$ of $G_1$. Each eigenvector $\phi_i^2$ is expressed as a linear combination of eigenvectors $S\phi_j^1$, and the combination coefficients are stored in row $i$ of $C$.

In Figure 3 we show several examples of matrix $C$ for different subgraphs. In the left side, the slanted-diagonal structure of the map between $\mathcal{M}$ and $\mathcal{N}$ is explained by an application of Weyl's law to 2-dimensional Riemannian manifolds, see (Rodolà et al., 2017, Eq. 9) and Appendix A.2. However, there is no theoretical counterpart to explain the map structure between $G_1$ and its subgraphs, due to the complete absence of metric information about the underlying surface: the eigenfunctions are computed *purely* from the graph connectivity. Yet, the diagonal structure is preserved even under rather dense removal of edges, suggesting deeper algebraic implications.

One might legitimately ask whether the presence of a structure in the maps of Figure 3 is due to the specific choice of the data, where the subgraphs derive from a 3D mesh (although the normalized

graph Laplacian dismisses any edge length information) and where the type of partiality resembles a neat 'cut' (although we also perform random edge removals). However, the same behavior is also observed with abstract graphs, as we show with CORA (McCallum et al., 2000) in Figure 4, and with the datasets PPI0 (Hamilton et al., 2017), Amazon Photo (McAuley et al., 2015) and Amazon Computer (McAuley et al., 2015) in Figure 7 of the Appendix.

To explain with an example how the structure of $C$ relates to the graph eigenspaces, consider the example of the Minnesota graph in Figure 1. Suppose we map the full graph to its permuted version (**i**). In this case, the two graphs have the same eigenspaces due to the permutation equivariance of Laplacian eigenvectors. Thus, matrix $C$ is diagonal with $\pm 1$ along the diagonal, because $c_{ij} = 0$ for $i \neq j$ (due to orthogonality of the eigenvectors), while $c_{ii} =$

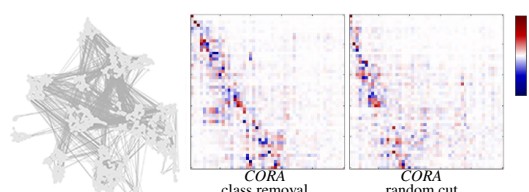

CORA
class removal

CORA
random cut

Figure 4: Functional maps between CORA and two different subgraphs.

$\pm 1$ (due to the sign ambiguity of the eigenvectors). In the case of repeated eigenvalues, one may observe small blocks of coefficients along the diagonal due to the non-uniqueness of the choice of the eigenvectors spanning high-dimensional eigenspaces. When we map the full graph to its subgraph (**ii**), the two graphs have partially similar eigenspaces, meaning that the inner products between $\phi_i^2$ and $S\phi_j^1$ tend to be close to zero and close to $\pm 1$, but not exactly equal. The matrix $C$ has a *sparse* structure but is not necessarily diagonal. This is because the eigenvectors on the subgraph correlate with those of the full graph at different indices $i \neq j$ – unlike the full-to-full case, where the correspondence happens at $i = j$.

***Remark (map structure):*** The functional map matrices are not necessarily diagonal, but may present a different sparsity structure which depends on the particular graph and subgraph.

As we will show in Section 5, the presence of a sparse structure in the functional map matrix $C$ works as a regularizer, in all those cases where the map is not given but must be estimated from the graph data.

### 4.4 NON-ISOMORPHIC SUBGRAPHS

In many practical settings, there are cases where the subgraph $G_2$ is contained in the bigger graph $G_1$ only up to some topological alterations; for example, in the graph learning literature, topological perturbations frequently occur due to noise in the data, or are explicitly obtained by rewiring operations (Chamberlain et al., 2021) or adversarial attacks (Jin et al., 2021) among others.

In Figure 1, we show the functional map between Minnesota and a subgraph after rewiring (**iii**). We still observe a correspondence between the eigenvectors of the full graph and those of the subgraph. The functional map has a sparse pattern, but it loosens up as the topological modifications increase. In Section 5.1, we further investigate this property and show the robustness of the functional representation to local topological changes.

***Remark (topological changes):*** The harder case, where there is partiality *in conjunction with* topological changes, still manifests a sparse structure in the coefficient matrix.

All the remarks so far directly depend on graph connectivity, and it is hard to find analogies for smooth surfaces. We conjecture that local topological transformations of a graph, while they can certainly induce strong transformations of *some* of its Laplacian eigenspaces (similar to single-point perturbations on planar manifolds, see Filoche & Mayboroda (2012)), are less likely to distort *all* the eigenspaces at once. This way, the functional map matrix tends to maintain its global structure intact, and exhibits local perturbations.

## 5 EMPIRICAL RESULTS AND ANALYSIS

In this section we validate the claims made in the previous sections with additional qualitative and numerical results. More experiments and details of the used datasets can be found in Appendix B.

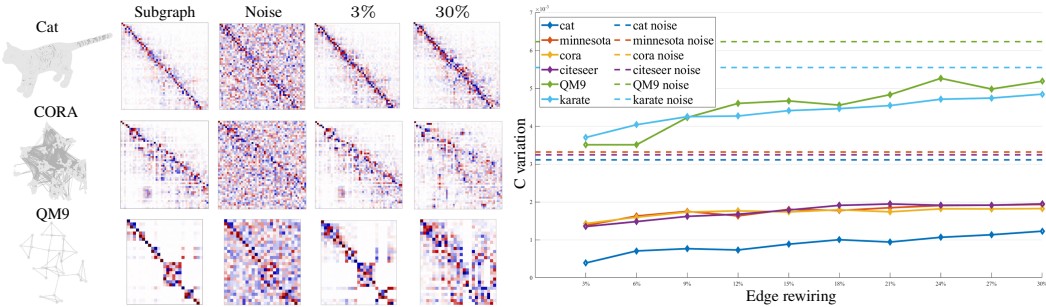

Figure 5: Robustness of the map to the simultaneous action of partiality and rewiring of the subgraph. We compare the addition of gaussian noise ($\mu = 0$;$\sigma = 0.2$) with the impact of increasing rewiring (from 3% to 30% of the total number of edges) on the functional map $C$ of size $50 \times 50$. On the left, we plot three graphs with their functional map: no rewiring (Subgraph), the addition of gaussian noise (Noise), 3% of edges rewired (3%), and 30% of edges rewired (30%). On the right, we plot the variation of $C$ at different percentages of rewiring (solid lines) and with the addition of noise (dashed lines) for each graph.

## 5.1 ROBUSTNESS TO REWIRING

In section 4.4, we claimed that the functional representation is robust to changes in graph connectivity. For this to be true, we expect that small changes in graph connectivity lead to small changes in the matrix coefficients. See Appendix B.2 for the formal definition.

In Figure 5, we evaluate the changes of the functional map at increasing percentages of rewiring of a subgraph. We consider six graphs and compute a subgraph from each one. Then, we apply small incremental changes to the topology of the subgraphs, with increments of 3% of the total number of edges; the changes are performed by removing and adding random edges, obtaining new subgraphs $G_i$. The plot on the right shows how much the functional map representation is affected by the increasing topological changes compared to adding Gaussian noise. In all the cases, the rewiring produces less variation in the functional map than adding Gaussian noise. In particular, the functional representation is more robust on larger graphs, such as cat (10000 nodes) or citeseer (2120 nodes), while on smaller graphs such as QM9 (29 nodes) and Karate (34 nodes), removing or adding an edge has a more significant impact. This observation demonstrates the effectiveness of the spectral representation, especially on larger graphs. In Appendix B.2, we show the complete qualitative analysis; while in Appendix C.2 we push this experiment to stronger rewiring.

## 5.2 SIGNAL TRANSFER

Within a graph, nodes may often come with numerical or vector-valued attributes, for instance, encoding molecular properties in PPIs, user identities in social networks, or positional encodings to better distinguish and characterize nodes. We can model such data as a collection of functions $f : V_1 \to \mathbb{R}$ that map each node of $G_1$ to a real value. Recent works (Brüel-Gabrielsson et al., 2022) have demonstrated that transferring the positional encoding from a graph to its rewired version can improve GNN performance. The functional map $C$ allows us to transfer these functions from a graph $G_1$ to a subgraph $G_2$ without requiring the explicit computation of a node-to-node correspondence. For each function $f$, it is sufficient to project $f$ onto the Laplacian eigenfunctions of $G_1$, apply the linear transformation $C$ to the obtained coefficients, and finally reconstruct the signal on the target graph $G_2$ as a linear combination of its Laplacian eigenfunctions. The described procedure is implemented by the simple formula:

$$\hat{g} = \Phi_2 C \Phi_1^\top f, \tag{2}$$

where the function $\hat{g} : V_2 \to \mathbb{R}$ is the transfer of $f$ to $G_2$. Motivated by the results from Brüel-Gabrielsson et al. (2022), we leverage this property of the spectral representation to transfer the positional encoding computed on a graph to its subgraphs. In Figure 6, we analyze the functional map transfer performance while increasing the number of eigenfunctions used for the map representation. We consider pairs composed of the original graph and a series of subgraphs extracted according to a semantic criterion, e.g., nodes belonging to the same class or nodes connected by the

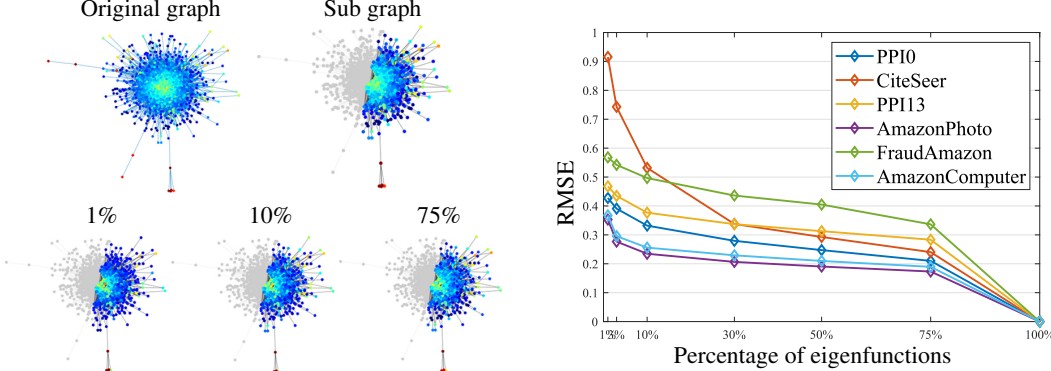

Figure 6: RMSE obtained by transferring positional encodings (PE) using the functional map with an increasing amount of eigenfunctions. On the left, we show a qualitative example of signal transfer on PPI0. The first row shows the full graph and the partial graph, with the PE plotted on top. The bottom row shows the results of signal transfer with different percentage of eigenfunctions. On the right, we plot the RMSE at increasing percentages of eigenfunctions.

same edge type. We transfer the Random Walk Positional Encoding (Dwivedi et al., 2022) computed on the full graphs to the subgraphs. We normalize each dimension of the node features of the original graph to exhibit zero mean and unitary standard deviation throughout all the nodes, and then transfer this signal through Equation 2. In Figure 6, we can see how the Root Mean Squared Error between the functional map and the ground truth transfer (we refer to Appendix B.3 for more details on the evaluation) decreases as the number of eigenfunctions increases. In particular, the error rapidly decreases at 10%-30% of eigenvectors. This behavior demonstrates that using a compact representation with few eigenvectors can approximate the signal well. The qualitative examples on the left of Figure 6 portray the transferred signal on PPI0. Already at 1% of the eigenfunctions the transfer reaches a good approximation, while at 75% it is almost identical. In Appendix A.3, we show more experiments with different number of eigenfunctions.

Suppose the graph and subgraphs are equipped with features independent of topological changes (like in Citeseer). In that case, we can compute the functional map $C$ from these features without needing ground truth correspondence a priori. To show the effectiveness of this alternative, in Appendix B.3, we show the results of signal transfer via an estimated $C$.

## 5.3 NODE-TO-NODE CORRESPONDENCE

Table 1: Comparison of the Mean Average Precision of different graph matching methods. We use $k = 50$ eigenfunctions for the functional map representation. We also report the percentage of eigenfunctions used w.r.t. the number of nodes of the full graph ($k\%$).

|  | Partiality | $k\%$ | IsoRank | FINAL | REGAL | PALE | GRASP | $FM_W$ | $FM_W$+ZM | $GT$ |
|---|---|---|---|---|---|---|---|---|---|---|
| Cat |  | 0.5 | $0.1 \pm 0.0$ | $0.2 \pm 0.0$ | $\mathbf{93} \pm 1.9$ | $6.8 \pm 0.4$ | $11 \pm 0.0$ | $68 \pm 14$ | $69 \pm 14$ | $92 \pm 3.9$ |
| Minnesota | patch | 1.9 | $0.2 \pm 0.1$ | $1 \pm 0.0$ | $87 \pm 3.3$ | $13 \pm 0.4$ | $18 \pm 2$ | $88 \pm 4$ | $89 \pm 3.7$ | $\mathbf{94} \pm 1.6$ |
| Cora |  | 2 | $0.5 \pm 0.0$ | $0.6 \pm 0.0$ | $54 \pm 3.5$ | $22 \pm 2.9$ | $6.8 \pm 1.8$ | $33 \pm 9.4$ | $34 \pm 9.5$ | $\mathbf{65} \pm 3.1$ |
| Cora | class | 2 | $0.4 \pm 0.0$ | $0.4 \pm 0.0$ | $55 \pm 3.4$ | $17 \pm 2$ | $51 \pm 22$ | $71 \pm 16$ | $60 \pm 21$ | $\mathbf{85} \pm 6.5$ |
| Douban | online-offline | 1.3 | 0.6 | 1.1 | $\mathbf{70}$ | 6 | 0.7 | 0.8 | 0.8 | 3.1 |

One of the advantages of the spectral representation is to reduce the NP-hard problem of finding node-to-node correspondences (usually formulated as a quadratic assignment problem (Loiola et al., 2007)) to the more tractable (polynomial) problem of finding the linear transformation between the reduced eigenbases of the graphs (at least under the reasonable assumption of smoothness of the sought correspondence, i.e., nearby nodes on the input graph are located nearby also on the target subgraph). Assuming a list of $m$ corresponding functions staked column-wise in two matrices $F_1 = \{f_1^{(1)} | \ldots | f_1^{(m)}\}$ and $F_2 = \{f_2^{(1)} | \ldots | f_2^{(m)}\}$, respectively defined on the node sets $V_1$ and $V_2$, the general functional map matching algorithm corresponds to the following minimization problem:

$$\arg\min_C \|C\Phi_1^\top F_1 - \Phi_2^\top F_2\|_2^2 + reg(C) \quad, \tag{3}$$

where $reg(C)$ is a regularization imposing desired properties on $C$ (e.g. sparsity, diagonal structure). The corresponding functions can be any consistent function between the two graphs, such as input features coming with the data.

In this experiment, we apply two functional map-based matching algorithms. We adopt the off-the-shelf partial functional map algorithm ($FM_W$) (Rodolà et al., 2017) and then the ZoomOut refinement (Melzi et al., 2019) ($FM_W + ZM$). We compare the spectral representation with the network alignment methods proposed in the benchmark (Trung et al., 2020): IsoRank (Singh et al., 2008), FINAL (Zhang & Tong, 2016), REGAL (Heimann et al., 2018) and PALE (Man et al., 2016). For all the methods, we use as input 50 landmark matches. For the two functional map methods we truncate the basis to the first 50 eigenvectors and use smooth indicator functions computed on the given landmarks as corresponding functions. We refer to the appendix B.4 for further details.

To investigate the degradation of the correspondence induced by the truncated eigenbasis, excluding errors generated by the matching algorithm, we also compute the correspondence starting from the ground-truth functional map (GT in the table).

We report in Table 1 the Mean Average Precision (MAP) defined as $\frac{1}{n} \sum_{i=1}^{n} \frac{1}{ra_i}$ where $ra_i$ is the rank (position) of positive matching node in the sequence of sorted candidates. For our evaluation we consider different graphs and subgraphs. We extract 10 partial graphs by considering k-hops subgraphs starting from random points. As we can see from the $GT$ column, the functional representation seems to preserve well the correspondence on these graphs, reaching the highest performance in almost all cases. Even if the functional map based methods leverage a more compact representation, their performance is satisfactory, being the best performing method in Minnesota and the second best in the other two datasets. Moreover, Appendix B.5 demonstrates that the functional map representation remains the fastest method in every graph. We note that for social graphs like CORA, characterized by few nodes with high degrees, the performance improves when the subgraph is semantically meaningful, obtained by removing all the nodes belonging to a specific class (fourth row). REGAL performs better since it is an ad-hoc algorithm for node-to-node correspondences on graphs, while our method is a more general solution that leverages a simple linear system. We find it remarkable that using a functional map as a representation without any fine-tuning works better in most cases, and are positive that this could lead to follow-up work for subgraph matching.

Finally, we test the functional representation on Douban (Wu et al., 2016), a real-world dataset composed of an online and offline version of the same social network. In this case, both the nodes and connectivity are very different between the two graphs, resulting in incompatible eigenbases making this scenario particularly challenging for a functional representation, as can also been observed by the low accuracy achieved using the $GT$ functional map. In Appendix C and A.3, we further investigate the performance of the functional representation at increasing partiality and number of eigenfunctions.

## 6  CONCLUSIONS

The spectral representation of functional maps for encoding graph and subgraph correspondences lends itself well to several applications, and we anticipate that it will be a useful addition to the graph learning toolset. Among the promising directions that we aim to explore are the definition of novel positional encodings for graphs robust to partiality transformations and to graph rewiring, and the application of functional mapping to more abstract structures such as learned graph embeddings.

Further, while in this paper we showed extensive evidence that the spectral map representation bears a special structure depending on the type of partiality, currently we have not taken full advantage of this structure. When the task at hand requires seeking for the subgraph alignment, i.e. whenever the map is unknown, it may be possible to design stronger regularizers to induce sparsity in the matrix representation of the map. This is quite different from the better known setting of 3D surfaces, where this sparse structure is typically just diagonal or slanted-diagonal.

In the light of the increasing interest of the graph learning community toward spectral techniques, adopting a spectral representation for maps between graphs is a natural next step; it is simple to adopt, easy to manipulate, and memory-efficient, and has the potential to become a fundamental ingredient in spectral graph learning pipelines in the near future.

## 7 REPRODUCIBILITY STATEMENT

All the code and generated data will be publicly released upon acceptance. In the supplementary material, we have provided an anonymous sample of the MATLAB code for reproducibility. In Section 5, we have specified the parameters used in the experiments, such as the number of eigenvectors, the error formulation or the probe functions. In particular, Appendix B focuses on clarifying our experiments' details and presenting the dataset used throughout our work.

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

## A INTERPRETATION OF THE FUNCTIONAL MAP MATRIX

### A.1 MATRIX REPRESENTATION

In the case of graphs, the functional map matrix is simply written as:

$$C = \Phi_2^\top S \Phi_1 \,, \tag{4}$$

where $\Phi_1 \in \mathbb{R}^{|V_1| \times k}, \Phi_2 \in \mathbb{R}^{|V_2| \times k}$ contain the first $k$ eigenvectors of the symmetrically normalized graph Laplacians of $G_1$ and $G_2$ respectively, and $S \in \mathbb{R}^{|V_1| \times |V_2|}$ is a matrix encoding the node-to-node correspondence. Note that differently from the case of surface meshes, in Equation 4 we write $\Phi_2^\top$ instead of using the pseudo-inverse $\Phi_2^\dagger$; this is due to the fact that the graph Laplacian eigenvectors are orthogonal with respect to the standard dot product, i.e., $\Phi_2^\top \Phi_2 = I$ and $\Phi_1^\top \Phi_1 = I$. This makes the matrix $C$ easy to compute by simple matrix multiplication.

### A.2 COMPARISON WITH SMOOTH SURFACES

In the case of smooth surfaces, it has been shown (Rodolà et al., 2017) that the sparsity pattern of matrix $C$ can be well approximated by a simple formula. Given a surface $\mathcal{M}$ and an isometric part $\mathcal{N}$, the matrix $C$ is approximately diagonal, with diagonal angle $\alpha$ proportional to the ratio of surface areas:

$$\alpha \sim \frac{\text{Area}(\mathcal{N})}{\text{Area}(\mathcal{M})} \,. \tag{5}$$

As a a special case, full-to-full isometric shape matching yields a diagonal matrix $C$, since $\text{Area}(\mathcal{N}) = \text{Area}(\mathcal{M})$. This result comes directly from an application of Weyl's asymptotic law for Laplacian eigenvalues of smooth manifolds (Weyl, 1911), which relates the eigenvalue growth to the surface area of the manifold via the relation:

$$\lambda_\ell \sim \frac{(2\pi)^2}{\text{Area}(\mathcal{M})^{2/d}} \ell^{2/d}, \qquad \ell \to \infty \tag{6}$$

where $d$ is the dimension of the manifold ($d = 2$ for surfaces). We refer to (Rodolà et al., 2017, Eq. 9) for additional details pertaining surfaces.

However, Weyl's law (Equation 6) does *not* hold for graphs, since there is not a well-defined notion of "area" of a graph. In fact, when we work with graphs and subgraphs, we observe that matrix $C$ does not necessarily follow a diagonal pattern. More general sparse structures are observed in the coefficients of $C$, but an explanation rooted in differential geometry is not readily available.

In Figure 7, we report additional examples with large abstract graphs undergoing partiality transformations, showing that clear patterns appear rather consistently across different datasets.

Based on these observations, we believe there is an intriguing theoretical gap between what has been observed in the case of smooth manifolds, and what we report for graphs in this paper. In the former case, a geometric explanation has been proposed in the literature. In the latter case, empirical evidence yields similar results, yet it seems to be a purely algebraic phenomenon that remains to be addressed.

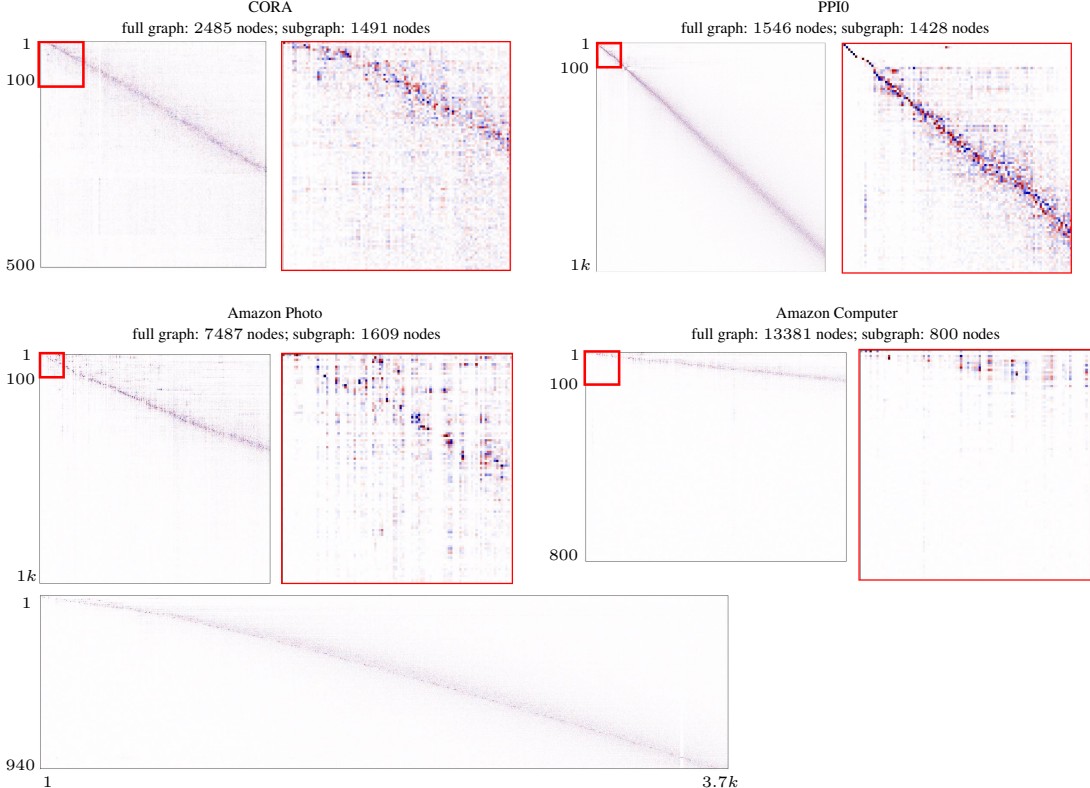

Figure 7: Functional maps computed over abstract graphs from 4 different datasets (CORA (McCallum et al., 2000), PPI0 (Hamilton et al., 2017), Amazon Photo (McAuley et al., 2015) and Amazon Computer (McAuley et al., 2015)), showing a clear pattern in all cases. For each dataset, we compute the functional map matrix $C$ between the complete graph and a subgraph; the subgraph is obtained according to a semantic criterion depending on the dataset, e.g., for Amazon Photo, by considering the subgraph of nodes belonging to the same product category. For each functional map matrix $C$, we also show a zoom-in (framed in red). All the matrices are sparse, and have a clean structure that in some cases approximates a slanted diagonal. The wide matrix on the bottom is computed on Amazon Photo (using a different subgraph than the one used in the example above it), and shows that the sparse behavior is maintained throughout the entire spectrum.

## A.3 NUMBER OF EIGENVECTORS

Given two graphs $G_1$ and $G_2$ with $m$ and $n$ nodes respectively, the node-to-node map $S$ has size $n \times m$, thus scaling quadratically with the number of nodes.

By contrast, matrix $C$ as defined in Equation 4 has dimensions that only depend on the number of Laplacian eigenvectors encoded in the matrices $\Phi_1, \Phi_2$. If one chooses the first $k_1 \ll m$ Laplacian eigenvectors for $G_1$ and the first $k_2 \ll n$ Laplacian eigenvectors for $G_2$, the size of $C$ is $k_2 \times k_1$. Observe that $C$ is rectangular in general, but can be made square by choosing $k_1 = k_2$ if so desired.

The experiments in Figure 6 and 8 show that as the number of eigenvectors increases, the performance also increases. In particular, Figure 8 demonstrates that, in most of the cases, a low percentage of eigenvectors (about 5%) suffices to retrieve a good node-to-node correspondence; while at 50% of the eigenvectors on all graphs the error is above 90%. As a general guideline, in this paper we typically use $k = 20 \sim 50$ for a graph with 1000 nodes, leading to an especially compact representation $C$.

## A.4 REGULARIZING BEHAVIOR

Using $k \ll n$ eigenvectors in the construction of $C$ has a regularizing effect on the map, akin to a low-pass filtering of the correspondence.

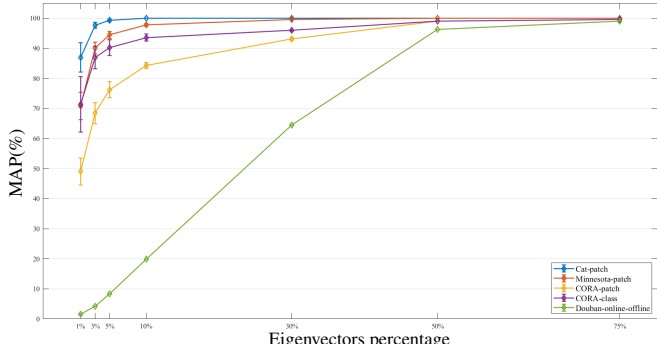

Figure 8: MAP(%) of the correspondence on different datasets at increasing number of eigenvectors (expressed as percentages, growing from $1\%$ to $75\%$). The correspondences are obtained from ground-truth functional maps.

In particular, when we use *all* the eigenvectors $\Phi_1$ and $\Phi_2$ to construct $C$, Equation 4 corresponds to an orthogonal change of basis; therefore, the representations $S$ and $C$ are equivalent and have the same dimensions. Truncating the bases to the first $k_1$ and $k_2$ eigenvectors, as described in Appendix A.3, yields a low-rank approximation $C \approx S$. In signal processing terms, we see the matrix $C$ as a *band-limited* representation of the node-to-node correspondence $S$.

The regularizing effect is desirable in many cases, but is traded off for a loss in accuracy if a precise node-to-node correspondence is desired. On the one hand, if the map $C$ is used to transfer a smooth signal (e.g. node-wise features like spectral positional encodings or carrying semantic information depending on the data), then the loss in accuracy is negligible, since Laplacian eigenvectors are optimal for representing smooth signals (Aflalo et al., 2015); on the other hand, transferring non-smooth signals via a small $C$ has the effect of filtering out the high frequencies. If high frequencies are desired, it is often sufficient to just increase the values of $k_1, k_2$, leading to a bigger matrix $C$.

## A.5 CHOICE OF LAPLACE OPERATOR

A functional map can be computed from the eigenbasis of any linear operator. In this paper we use the symmetrically normalized graph Laplacian $\mathcal{L} = I - D^{\frac{1}{2}} A D^{\frac{1}{2}}$. A valid alternative is the standard Laplacian $L = D - A$, which shows similar behavior to the normalized counterpart. At a practical level, we observed that the Laplacian $L$ suffers from more problems of high multiplicity at lower frequencies, see Figure 9.

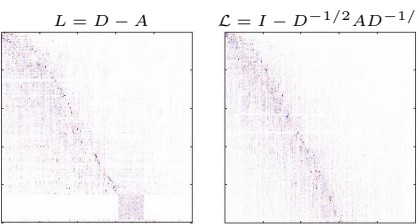

Figure 9: Functional map computed with two different Laplacians between the CORA graph and its subgraph.

In the special case where the graph is constructed on top of a point cloud sampled from a (possibly high-dimensional) manifold $\mathcal{M}$, it has been shown that the eigenvectors of the normalized graph Laplacian converge to the eigenfunctions of the Laplace-Beltrami operator on $\mathcal{M}$ (Belkin & Niyogi, 2006). However, as discussed in Appendix A.2, our case is more general. We consider generic abstract graphs without an explicit underlying manifold, i.e. we do not construct our graphs from input point clouds. Further, in Belkin & Niyogi (2006) it is assumed that $\mathcal{M}$ is a compact infinitely differentiable Riemannian submanifold of $\mathbb{R}^d$ *without* boundary, meaning that partiality transformations, which are the main focus of this paper, are not considered.

## B DATASET AND IMPLEMENTATION DETAILS

In this section we report additional details about the experimental setup used in the main manuscript.

## B.1 DATASETS

In Table 2 we sum up the main statistics across all the datasets and benchmarks used in our experiments. In addition to number of nodes, number of edges, graph diameter and average node degree, in the table we also report the application domain of each dataset, the task where they are used, the type and number of node-wise features (where used). Since PPI and QM9 are collections of graphs, we used only a subset. In particular, from the PPI dataset we used the first and fourteenth graphs (specified with 0 and 13 in the experiments). The Cat graph is derived from the corresponding mesh of the SHREC'16 Partial Deformable Shapes benchmark (Cosmo et al., 2016b).

Table 2: Summary of statistics about the datasets used in our experiments.

| Dataset | Nodes | Edges | Diameter | Average degree | Domain | Task | Features | Number of features |
|---|---|---|---|---|---|---|---|---|
| QM9 (Klicpera et al., 2020) | 29 | 47 | 6 | 3.24 | Chemistry | Graph regression | - | - |
| Karate (Zachary, 1977) | 34 | 78 | 5 | 4.59 | Social networks | Node classification | - | - |
| PPI 0 (Hamilton et al., 2017) | 1546 | 17699 | 8 | 21.90 | Chemistry | Graph regression | Gene attributes | 50 |
| Citeseer (Giles et al., 1998) | 2120 | 3731 | 28 | 3.50 | Citation networks | Node classification | Bag-of-Words | 3703 |
| Cora (McCallum et al., 2000) | 2485 | 5069 | 19 | 4.08 | Citation networks | Node classification | - | - |
| Minnesota | 2635 | 3298 | 98 | 2.5 | Roadmap | - | - | - |
| PPI 13 (Hamilton et al., 2017) | 3480 | 56857 | 8 | 31.68 | Chemistry | Graph regression | Gene attributes | 50 |
| Douban (Wu et al., 2016) | 3906 | 8164 | 13 | 4.18 | Social networks | Network alignment | - | - |
| Amazon Photo (McAuley et al., 2015) | 7487 | 119044 | 11 | 31.80 | Co-purchase | Node classification | Bag-of-Words | 745 |
| Cat (Cosmo et al., 2016a) | 10000 | 19940 | 86 | 5.99 | Geometry processing | Shape matching | - | - |
| FraudAmazon (Zhang et al., 2020) | 11944 | 4417576 | 4 | 739.71 | Product reviews | Fraud detection | Bag-of-Words | 25 |
| Amazon Computer (McAuley et al., 2015) | 13381 | 245778 | 10 | 36.74 | Co-purchase | Node classification | Bag-of-Words | 767 |

## B.2 ROBUSTNESS TO REWIRING

In this Section, we formally define the connectivity changes and functional map robustness used in Section 5.1. Given two graphs $G = (V, E)$ and $G' = (V', E')$, we measure the amount of change from $G$ to $G'$ as the (minimum) number of edits needed to transform $E$ to $E'$, divided by $|E|$: $\frac{(|E - E'| + |E' - E|)}{|E|}$. In our experiments, we consider small changes in the graph connectivity as a perturbation of 3% of the edges. The rewiring operation that we applied to the graphs consists of the deletion or addition of the same amount of edges.

We define the difference between the functional map $C$ and $C'$ as $\|C - C'\|_F^2$. Note that there is ambiguity in the sign of the eigenfunctions of $C'$; to factor it out from the error computation, we use the sign that minimizes the error.

In Figure 10 we show the functional maps generated from the experiment in Figure 5. Figure 10a shows the functional map between the full and partial graphs from 0% to 30% of rewiring; Figure 10b shows the variation in the functional representation between the non-rewired case and the different percentages of rewiring.

## B.3 SIGNAL TRANSFER

We evaluate the fidelity of the transferred signal with the Root Mean Squared Error between the transferred signal $\hat{g}$ and the ground truth signal $g$ (obtained via the ground truth node-to-node correspondence):

$$RMSE = \sqrt{\frac{1}{n}\sum_{i}^{n}(g(i) - \hat{g}(i))^2},\qquad(7)$$

where $n$ is the number of nodes in the subgraph.

As anticipated in the main manuscript, the functional map C can also be computed from features independent of topological changes with Equation 8. On Citeseer, the node features are Bag-of-Words that identify each node, therefore they should not change under perturbations of the graph topology. We use as probe functions the features defined on each node and compute a map $C$ using 50 eigenfunctions without any regularizer or refinement to

Table 3: RMSE of the signal transfer computed with two functional maps obtained with different methods.

| Subgraph | $C_{gt}$ | $C$ |
|---|---|---|
| 1 | 0.68 | 0.85 |
| 2 | 0.96 | 0.97 |
| 3 | 0.60 | 0.63 |
| 4 | 0.81 | 0.83 |
| 5 | 0.72 | 0.77 |
| Mean | 0.75 | 0.81 |

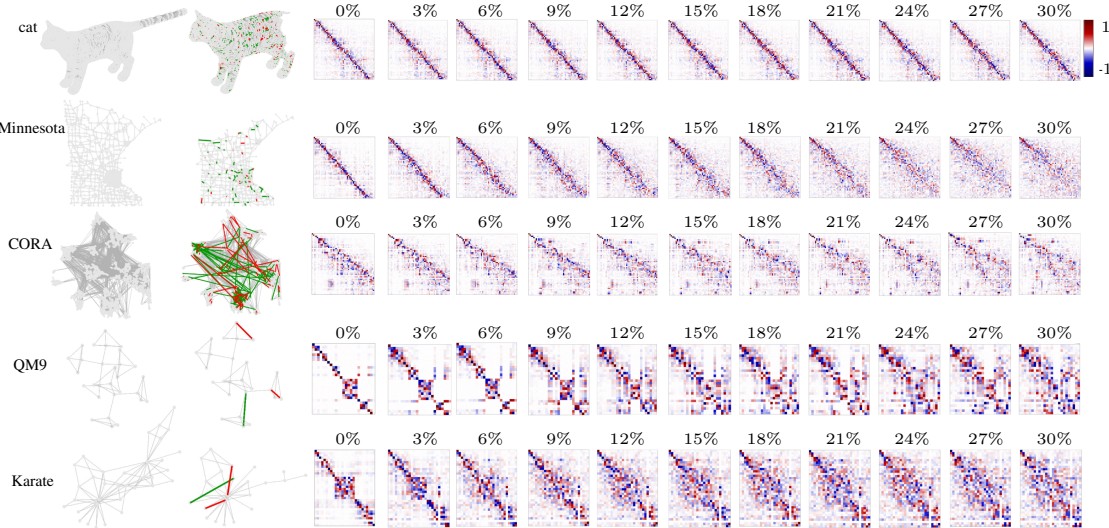

(a) The plotted matrices represent the functional map between the full and partial graphs from 0% to 30% of rewiring, showing the effect of rewiring on the functional map structure.

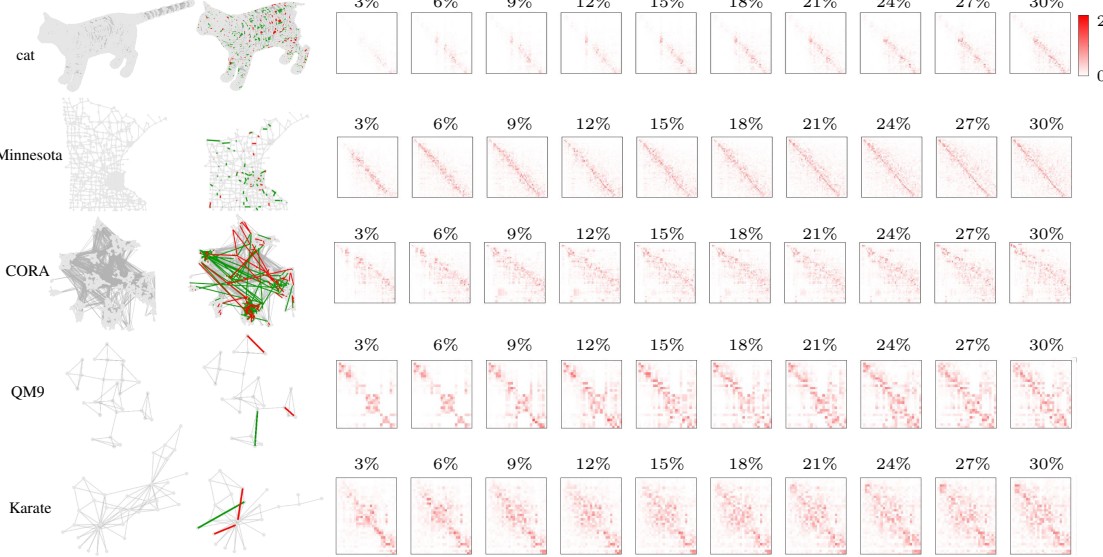

(b) The plotted matrices encode the element-wise error of the functional map after the topological perturbations. Error is encoded as color, growing from white to red.

Figure 10: Robustness of the map to the simultaneous action of partiality and rewiring of the subgraph. The rewiring operations are increasingly **stronger**, with increments of 3% of the total number of edges (starting from 3% and reaching 30%). The second column shows one representative example (per dataset) of such topological modifications, depicting the added edges in green, and the removed edges in red. The plotted matrices represent the functional map after the topological perturbations, showing the effect of rewiring on the functional map structure.

highlight the contribution of the features. We compare the results of this functional map $C$ with the ones of a map $C_{gt}$ obtained from the ground truth correspondences (Equation 4). Each row of Table 3 represents the RMSE of a different subgraph, while the bottom row reports the mean. The results show that even if we do not have a ground truth correspondence between two graphs, the functional map $C$ can transfer signals with a good approximation.

### B.4 NODE-TO-NODE MAP INFERENCE

As stated in the main manuscript, to test the effectiveness of the spectral representation when seeking a node-to-node correspondence matrix, we adopt the off-the-shelf partial functional maps algorithm ($FM_W$) (Rodolà et al., 2017) and then refine the correspondence with ZoomOut (Melzi et al., 2019) ($FM_W + ZM$). These algorithms are not specifically designed for graph matching, but can work with generic spectral representations, which is one of their main benefits.

We implemented the $FM_W$ optimization in MATLAB through the manopt package (Boumal et al., 2013). We considered a minimization problem with the form:

$$C = \operatorname*{arg\,min}_{C \in \mathbb{R}^{k \times k}} \|CA - B\|_{2,1} + \mathcal{R}(C)\,, \tag{8}$$

where $A$ and $B$ are the coefficients in the Laplacian basis for a set of corresponding probe functions, while $\mathcal{R}(C)$ is some additional regularizer. As probe functions we computed smooth $\delta$ functions for the given set of landmark matches. For all the methods that accepted nodes features (FINAL, REGAL, GRASP) as input, we used the same functions as nodes features. $\mathcal{R}(C)$ fosters additional structure to $C$ and is defined as:

$$\mathcal{R}(C) = \mu_1 \|C \odot W\|_2^2 + \mu_2 \sum_{\ell \neq h} (C^\top C)_{\ell,h}^2 + \mu_3 \sum_{\ell} ((C^\top C)_{\ell,\ell} - d_\ell)^2\,, \tag{9}$$

where each term has its weight $\mu_1$, $\mu_2$ and $\mu_3$. $W$ is a mask matrix that acts through element-wise multiplication $\odot$ and encodes the relation between the eigenvalues of the two shapes, which approximates the slanted-diagonal structure of $C$ induced by the partiality. In particular, we computed $W$ with the complex resolvent method proposed in Ren et al. (2020b). The term weighted by $\mu_2$ promotes orthogonality of the map by penalizing the off-diagonal entries of $C^\top C$. Finally $d_\ell \in \{0, 1\}$ $\forall \ell$; the entries equal to 1 represent which singular values of $C$ are expected to be non-zero. A refinement, similar to the iterative closest point algorithm (Besl & McKay, 1992) in the space of the coefficients, is then applied to the matrix $C$. As a final step, the spectral refinement approach of ZoomOut (Melzi et al., 2019) is applied to the computed $C$. Given a map of size $50 \times 50$ as input, we apply ZoomOut to its $37 \times 37$ sub-matrix and get back a refined matrix $C_{ZM}$ of size $50 \times 50$.

The ground-truth functional map ($GT$) is obtained through Equation 4 where $S$ is the ground-truth correspondence matrix.

All the methods we compared to in Section 5.3 of the main paper were taken from the public benchmark (Trung et al., 2020). To run the experiments, we used the standard parameters suggested in their code. As node features, we used the same probe functions defined from the landmarks as in $FM_W$. PALE was trained on the ground-truth correspondences given by the 50 landmarks.

### B.5 COMPUTATIONAL TIME

In Table 4, we report the optimization time needed to compute the correspondences of Table 1 from the main paper. For functional map-based matching algorithms ($FM_W$, $FM_W + ZM$) we did not consider the computation of the eigenvectors since it is an offline cost as eigenvectors can be pre-computed once and re-used. For $FM_W + ZM$, we only report the computational time of ZoomOut refinement starting from an input functional map. All the experiments were performed on a Intel(R) Core(TM) i7-9700K CPU @ 3.60GHz.

Table 4: Comparison of the computational time (in seconds) needed by different methods to compute a node-to-node correspondence, across several datasets. Spectral based methods (last two columns) are the most efficient.

| | Partiality | IsoRank | FINAL | REGAL | PALE | GRASP | $FM_W$ | $FM_W$ +ZM |
|---|---|---|---|---|---|---|---|---|
| Cat | | $270.44 \pm 9.84$ | $156.49 \pm 2.73$ | $36.78 \pm 0.86$ | $3586.4 \pm 109.5$ | $1057.7 \pm 100.59$ | $\mathbf{20.73 \pm 1.67}$ | $\mathbf{5.45 \pm 0.28}$ |
| Minnesota | patch | $32.18 \pm 3.94$ | $10.21 \pm 0.15$ | $7.08 \pm 0.07$ | $145.72 \pm 9.14$ | $648.43 \pm 267.5$ | $\mathbf{2.83 \pm 0.66}$ | $\mathbf{0.33 \pm 0.03}$ |
| Cora | | $31.54 \pm 3.1$ | $10.48 \pm 0.55$ | $7.33 \pm 0.15$ | $211.99 \pm 1.87$ | $198.22 \pm 148.3$ | $\mathbf{2.89 \pm 0.42}$ | $\mathbf{0.39 \pm 0.4}$ |
| Cora | class | $14.92 \pm 2.4$ | $51.61 \pm 10.2$ | $10.92 \pm 0.9$ | $434.13 \pm 35.6$ | $118.61 \pm 72$ | $\mathbf{1.96 \pm 0.1}$ | $\mathbf{0.38 \pm 0.02}$ |
| Douban | online-offline | $11.54$ | $27.07$ | $10.41$ | $337.56$ | $16.32$ | $\mathbf{4.43}$ | $\mathbf{0.03}$ |

(a) Patch

(b) Holes

Figure 11: MAP(%) of the correspondence on different datasets at decreasing size of the subgraph (expressed as percentages, decreasing from $90\%$ to $50\%$) with two types of partiality: patch (left) and holes (right). The correspondences are obtained from ground-truth functional maps.

## C   ADDITIONAL EXPERIMENTS

### C.1   PARTIALITY PERCENTAGE

The aim of this experiment is to study how much the performance of the functional representation for a correspondence task degrades with different levels and types of partiality.

In Figure 11, we perform an experiment similar to the one in Section 5.3 of the main paper and test how the MAP of the correspondence changes when we consider subgraphs of different dimensions. In particular, we test 2 partialities: patch and holes. The former is obtained by expanding a neighborhood from a given random node until a certain number of nodes is reached. The latter is obtained by removing random nodes with their immediate neighborhood. The main difference is in the sparsity of the two partialities. The first keeps a cluster of nodes without changing the inner connectivity of the graph; the latter is sparser in the sense that it removes paths between nodes of the graph, changing the whole connectivity of the graph. In Figure 12, we plot the functional maps at different percentages of partiality of type patch (left) and holes (right).

### C.2   REWIRING

In the main paper and in Figure 10, we highlight how the functional map degrades when we apply a sequence of local rewiring operations. Here we consider a much stronger variation, namely, a *global* rewiring where we do not limit the topological perturbation to a local neighborhood. The results are reported in Figure 13.

Finally, in Figures 14 and 15, we show the functional maps under the action of pure rewiring perturbations, without any partiality involved.

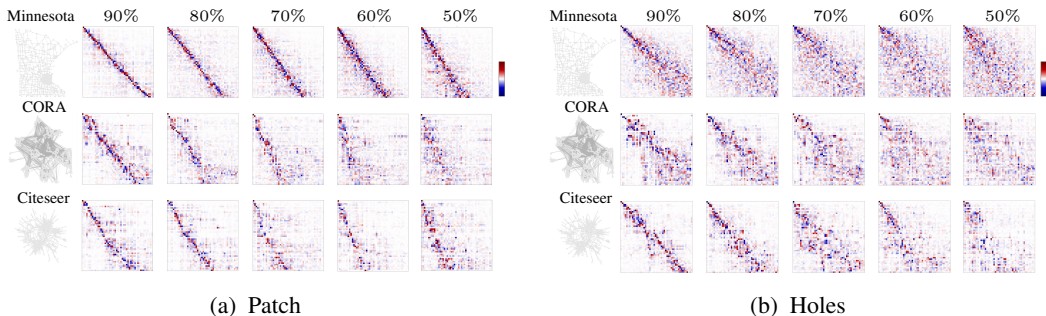

Figure 12: Robustness of the map to the action of two partialities: patch and holes. The partialities are increasingly wider, from a subgraph with 90% of the nodes to a subgraph with 50% of the nodes. As partiality increases, the functional map structure becomes less defined.

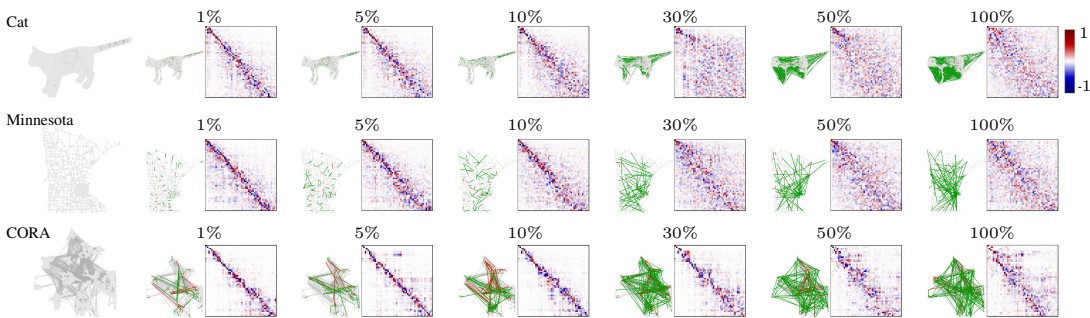

Figure 13: Robustness of the map to the simultaneous action of partiality and rewiring of the subgraph. The rewiring operations are increasingly **wider**, involving nodes and edges farther apart. The left column shows the full graph, while the pairs composed by a graph and a matrix show the rewiring operation and the resulting functional map, starting with a rewiring involving just nodes within 1% of the graph diameter (left) up to the full graph diameter (right).

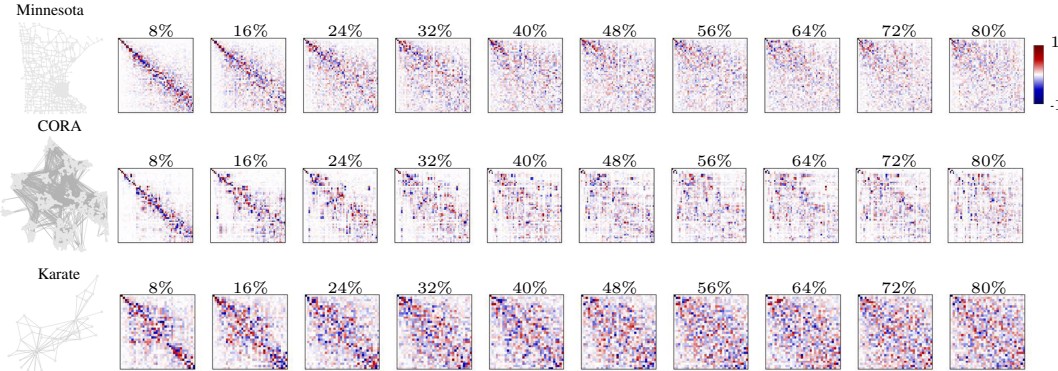

Figure 14: Robustness of the map to rewiring of the subgraph. The rewiring operations are increasingly stronger, with increments of 8% of the total number of edges (starting from 8% and reaching 80%). The plotted matrices represent the functional map after the topological perturbations, showing the effect of rewiring on the functional map structure.

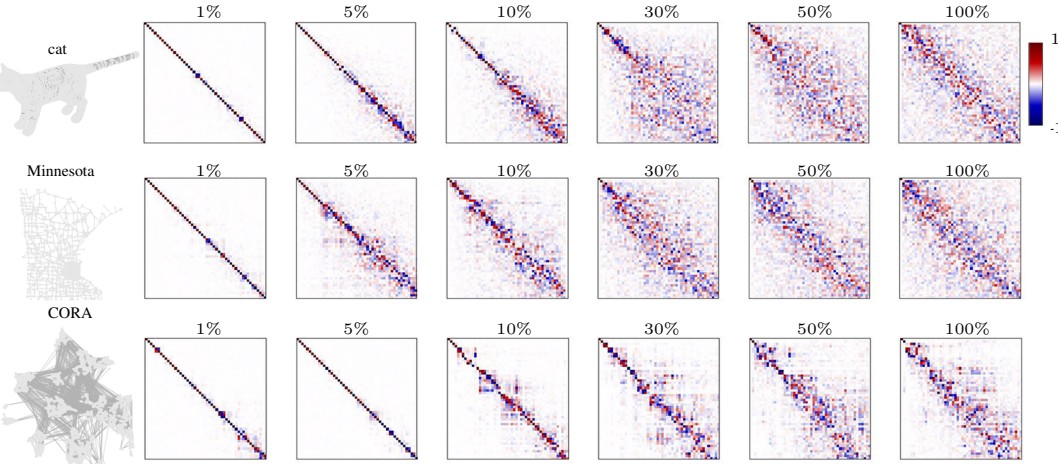

Figure 15: Robustness of the map to rewiring of the subgraph. The rewiring operations are increasingly wider, involving nodes and edges farther apart. The left column shows the full graph, while the others columns show the resulting functional map, starting with a rewiring involving just nodes within 1% of the graph diameter (left) up to the full graph diameter (right)

