# OpenReview forum: "Harnessing spectral representations for subgraph alignment"
_ICLR.cc/2023/Conference — Submitted to ICLR 2023_

### Official Review · Reviewer_rw4F · 2022-10-22

**Confidence:** 3
**Correctness:** 2
**Technical Novelty And Significance:** 2
**Empirical Novelty And Significance:** 2
**Recommendation:** 5

**Clarity, Quality, Novelty And Reproducibility:**

I am not fully understand the functional map part. I think they could also talk about what is the objective of the functional map. What is the meaning of C. And how would C be useful in terms of graph matching or what is the role of C in terms of graph matching not in terms of matching two eigen matrix.

**Strength And Weaknesses:**

The paper try to address the subgraph matching with eigen decomposition. And the whole paper based on a very strong assumption that the eigen information in subgraph is preserved compared with original graph.

This observation is quite empirical and without solid proof in this paper. Though the author conducted the extensive experiment but that does not exactly prove the same thing.

Frankly, I do agree in some cases, the subgraph would definitely preserve the information of original graph like in figure 2 of the paper. But for most of the cases, subgraph would be very different than the original graph. Thus it is very hard to do the matching/alignment. And make it difficult to evaluate this paper. Overall, I do not think this result is valid given the empirical observation. Thus I intend to give this paper a boarderline assessment.

I do think investing on some cornor case, where the eigen information did not preserve the information would help improve this paper.




**Summary Of The Paper:**

This paper talk in detail about the subgraph alignment problem. Which they assume the graph laplacian of the graph is well preserved by the subgraphs. There are some good merit of the paper in turns of experiments. But overall, I find the paper quit confusing. And the basis of the paper is too empirical and not justiable.

**Summary Of The Review:**

This paper try to tackle subgraph matching based on the empirical visualization. But without valid proof, the visualization is not very intuitive thus make it hard to evaluate the paper.

---

> ### Author Response · Authors · 2022-11-10
> **Further clarifications on corner cases**
>
> We thank the reviewer for the comments.
>
> We would like to ask for clarifications on an observation that is made in the review. In particular: *“I do think investing on some corner case, where the eigen information did not preserve the information would help improve this paper”*. We ask the reviewer to clarify on this part, by providing examples of more complicated cases where graph alignment can be tested. The reviewer also states *“for most of the cases, subgraph would be very different than the original graph”*; we agree with this statement, and precisely for this reason we also test the correlation between the eigenvectors of irregular graphs (such as CORA, Citeseer, Amazon Photo) and subgraphs where the former is a rewired version of a sub isomorphism. Even in these cases, we have shown that the functional map shows a sparse and compact representation. This is far from obvious and has not been reported in the literature before, to our knowledge. We have also demonstrated examples of signal transfer and graph matching between these kinds of graphs (see Sections 5.2 and 5.3).

---

### Official Review · Reviewer_BSxT · 2022-10-24

**Confidence:** 4
**Correctness:** 2
**Technical Novelty And Significance:** 2
**Empirical Novelty And Significance:** 2
**Recommendation:** 3

**Clarity, Quality, Novelty And Reproducibility:**

The article as it is currently written does not really make justice to the possible novelties, the methodological proposition  and the achieved results of the work. Although the idea is an interesting one, the way the article is written obfuscates the possible reach of the work.
I strongly recommend that this work is fully revised so as to write it in a more acceptable manner for the audience of this conference.

With improvements in quality and clarity of the main paper (and removal of useless parts), the originality of the work could be also  clearer.

**Strength And Weaknesses:**

Strength:

1- as mentioned above, the idea of using map correspondence is nice, and some insghts gained on the alignment between a graph and subgraphs (even non-isomorphic ones) are interesting. This does not appear to have be commented much.

2- the conceptual simplicity of the proposed method is interesting, meaning that the writing of eq. (4) appears almost like a method in itself.

Weaknesses :

1- The writing of the article is wrong for me : i) All the discussions in sections 1 and 2 are, for me, off for a scientific article. Some constatations in 1 are forced without actual any fact to support that (e.g. "has been largely overlooked" ; "common misconception" ? it could be mere lack of interest due to more relevant things to do) ; Section 2 is way too long to discusse node-to-node correspondence and functional correspondence while there is only scarce comparisons to these state-of-the-art methods in the text (only in Table 1, which is not that conclusive, see underneath)

2- On the other side: the interesting parts of the paper are mostlty in the Appendices.: A.1, A.3, A.4, most of B. This is disturbing to read comments in the main text that are not always useful (details in Section 2), founded (in 1), or with enough details to understand (all of Section 4 where there is no equation describing the proposed method). In the current state of presentation, I don't think this article is suited to the audience of the conference (interested in understading the proposed method and not only in looking at results and discussions).

3- the mentioned insight about alignment between a graph and subgraphs is not that surprising for people working witgh Graph Laplacians. Hence, the presentation of the results related to that should be dowtoned a bit.

4- the results in 5.3, about node-to-node correspondence, are somehow disappointing (especially w.r.t. the high hopes and stakes puty forward by the authors in Sec 1 to 4): on the task, REGAL performs way better; the proposed method is significantly further away from Ground Truth.

5- I didn't find any point of  comparisons for results in 5.1 and 5.2. No SOTA to compare to ?




**Summary Of The Paper:**

The article discusses the adaptation of manifold alignement (as studied, e.g., 10 years ago in Osjanikov et al, 2012) to the setting of graphs. One relevant and nice idea is not to consider not only full graph alignment, but also subgraph alignments even in the non-isomorphic case. The authors show that the notion of functional map correspondance can be translated to graphs as looking at mixed eigenmaps connecting eigenvectors of the two considered domains (graphs). Then some empirical results follow.

**Summary Of The Review:**

For all the reasons above, my initial recommendation is to reject the paper, yet I would be happy to revise lmy judgement if the authors revise significantly the manner to present the work, so as to improve clarity and better put forward the methodological novelties that are now hidden in lengthy appendices and absent from the main text.

---

> ### Author Response · Authors · 2022-11-10
> **Further references on well known graph and subgraph alignment**
>
> We thank the reviewer for the comments.
>
> The reviewer stated that *“the mentioned insight about alignment between a graph and subgraphs is not that surprising for people working with Graph Laplacians”*. To our knowledge, this is not the case, and the lack of such evidence is precisely one of the key points of our submission. In view of improving our work, we would like to ask for exact references where these alignments are presented as well known.

---

### Official Review · Reviewer_iFfS · 2022-10-25

**Confidence:** 4
**Correctness:** 2
**Technical Novelty And Significance:** 2
**Empirical Novelty And Significance:** 2
**Recommendation:** 3

**Clarity, Quality, Novelty And Reproducibility:**

The paper is written clearly. The paper lacks sufficient comparison with the techniques in the existing literature and rigor in the experiments to judge novelty. The observation that a correlation may exist between the eigenvectors of full graph and its sub-graphs is novel but the paper does not provide enough details on the scenarios where such a correlation may exist.

**Strength And Weaknesses:**

Strengths:

Alignment between graphs and their sub-graphs is an interesting problem. The paper provides convincing empirical evidence that a correspondence between the functional maps (defined by the eigenvectors) may exist between graphs and their sub-graphs even when the sub-graphs are localized.

Weaknesses:

1. **Related Work** : The comparison with existing works in the literature is critically lacking. There exists a rich literature on domain adaptation and meta learning where the contributions involve learning linear mapping to align data from two domains.

      *a. Sun, Baochen, Jiashi Feng, and Kate Saenko. "Return of frustratingly easy domain adaptation." Proceedings of the AAAI Conference on Artificial Intelligence. Vol. 30. No. 1. 2016.*

      *b. Yang, Yi, et al. "Data-Efficient Brain Connectome Analysis via Multi-Task Meta-Learning." arXiv preprint arXiv:2206.04486 (2022).*

     *c. Pilanci, Mehmet, and Elif Vural. "Domain adaptation on graphs by learning aligned graph bases." IEEE Transactions on Knowledge and Data Engineering (2020).*

    Moreover, inference over graph limit objects, such as graphons, relates to the setting of a subgraph constructed by eliminating nodes randomly all over the graph (such as in Fig. 3).

2. While the observation that the eigenvectors of a partial graph and the complete graph may correlate is not trivial, the experiments provide no concrete insight into how and why such correlation may exist. For instance, can you meaningfully quantify the similarity between the subgraph and graph sufficient for the correlation pattern in Fig. 2 to emerge? Are the datasets used in Fig. 6  and Fig. 8 low rank?

3. The paper focuses on learning only linear mappings whereas there seem to be no technical challenges to learn non-linear mappings between the functional maps.



**Summary Of The Paper:**

This paper proposes that a robust spectral alignment map may exist between graphs and partial graphs (which could be subgraphs in the original graph). The claims are driven by empirical observations on various real-world datasets.

**Summary Of The Review:**

Subgraph alignment is an interesting observation. However, the paper lacks in comparisons with existing literature. The experiments are not rigorous enough for a reader to draw meaningful insights that could be generalized beyond the datasets presented in this paper.

---

### Decision · Program_Chairs · 2023-01-20

**Decision:**

Reject

**Justification For Why Not Higher Score:**

All reviewers raised major concerns on this paper, including writing, technical methods and details, and results. Given the consistency and extents of these concerns, a reject is appropriate.

**Justification For Why Not Lower Score:**

NA

**Metareview: Summary, Strengths And Weaknesses:**

This paper proposes spectral methods for subgraph alignment. All reviewers raised major concerns on this paper, including writing, technical methods and details, and results. Given the consistency and extents of these concerns, a reject is appropriate.

**Summary Of Ac-Reviewer Meeting:**

NA